# Molecular and Morphological Profiling of Lung Cancer: A Foundation for “Next-Generation” Pathologists and Oncologists

**DOI:** 10.3390/cancers11050599

**Published:** 2019-04-29

**Authors:** Jumpei Kashima, Rui Kitadai, Yusuke Okuma

**Affiliations:** 1Department of Pathology, Tokyo Metropolitan Cancer and Infectious diseases Center Komagome Hospital, Tokyo 113-8677, Japan; jkashima-tky@umin.ac.jp; 2Department of Pathology, Graduate School of Medicine, The University of Tokyo, Tokyo 113-0033, Japan; 3Department of Thoracic Oncology and Respiratory Medicine, Tokyo Metropolitan Cancer and Infectious Diseases Center Komagome Hospital, Tokyo 113-8677, Japan; rkitadai@cick.jp; 4Department of Thoracic Oncology, National Cancer Center Hospital, Tokyo 104-0045, Japan

**Keywords:** histological subtype, molecular pathology, targeted therapy, lung cancer

## Abstract

The pathological diagnosis of lung cancer has largely been based on the morphological features observed microscopically. Recent innovations in molecular and genetic technology enable us to compare conventional histological classifications, protein expression status, and gene abnormalities. The introduction of The Cancer Genome Atlas (TCGA) project along with the widespread use of the next-generation sequencer (NGS) have facilitated access to enormous data regarding the molecular profiles of lung cancer. The World Health Organization classification of lung cancer, which was revised in 2015, is based on this progress in molecular pathology; moreover, immunohistochemistry has come to play a larger role in diagnosis. In this article, we focused on genetic and epigenetic abnormalities in non-small cell carcinoma (adenocarcinoma and squamous cell carcinoma), neuroendocrine tumor (including carcinoids, small cell carcinoma, and large cell neuroendocrine carcinoma), and carcinoma with rare histological subtypes. In addition, we summarize the therapeutic targeted reagents that are currently available and undergoing clinical trials. A good understanding of the morphological and molecular profiles will be necessary in routine practice when the NGS platform is widely used.

## 1. Introduction

In the present era of precision medicine, the paid cost and timewise availability of genome sequencing has made it possible to screen oncogenic genetic alterations and epigenetic abnormalities in a patient. The mortality rate of lung cancer is the highest among all cancers worldwide [1]. This condition has been treated using molecular targeted therapy since early days. Epidermal growth factor (EGFR) and anaplastic lymphoma kinase (ALK) inhibitors are some of the widely available targeted agents in non-small lung cancer (NSCLC), demonstrating dramatic clinical effects when compared to conventional cytotoxic chemotherapy. Needless to say, targetable treatments predispose specific gene alterations, and their prevalence differs according to the histology of the lung cancer. The Cancer Genome Atlas (TCGA) project and numerous other studies have shown molecular characteristics in each major histological subtype of lung cancer (Figure 1) [2].

In this article, we focused on the association between morphological classifications and molecular alterations of epithelial lung cancer. In addition, we have listed the targeted therapies that are currently available and may be used in future against lung cancers.

## 2. Adenocarcinomas and Squamous Cell Carcinomas

### 2.1. Morphological and Immunohistochemical Diagnosis

Adenocarcinoma (ADC) is classically defined as a cancer that presents in a glandular pattern or bears cytoplasmic mucus whereas, squamous cell carcinoma (SQC) is characterized by the presence of intercellular bridges and keratinization. These morphological characteristics are occasionally obscure due to the limitation of tissue availability, especially in advanced diseases. Such cases have been summarized as “large cell carcinoma” in the past. However, some therapeutic reagents (pemetrexed and bevacizumab) cannot be administrated in SQC patients due to the risk of hemorrhagic adverse events; moreover, targeted therapies according to background molecular alterations are different between ADC and SQC. Therefore, immunohistochemical analysis (IHC) should be utilized to distinguish between these two histological types in tiny tissue samples and/or poorly-differentiated NSCLC. TTF-1 and Napsin A are assumed as ADC markers, while p40 and Cytokeratin 5/6 are used as SQC markers.

### 2.2. Adenocarcinoma

#### 2.2.1. Histological Subtypes of ADC

Precursor lesions such as atypical adenomatous hyperplasia, adenocarcinoma in situ (mucinous and non-mucinous), and microinvasive ADC (≤0.5 cm invasive component; mucinous and non-mucinous) were newly introduced in the World Health Organization (WHO) classification of lung ADC (2015) [2]. Relationships between histological subtypes (lepidic, acinar, papillary, micropapillary and solid ADC) and prognosis have been evaluated in a previous study where lepidic was regarded as low-grade, acinar and papillary as intermediate-grade, and micropapillary and solid as high-grade [3]. Invasive mucinous ADC presents with abundant cytoplasmic and extracellular mucus, and harbors unique molecular profiles when compared with the non-mucinous ADC as described below. Other relatively rare subtypes of ADC (colloid ADC, fetal ADC, and enteric ADC) are also mentioned [2].

#### 2.2.2. Molecular Abnormalities in ADC Confirmed by TCGA

A comprehensive molecular analysis was performed against 230 lung ADCs in the TCGA project [3]. The histological breakdown was as follows: acinar (33.5%), solid (25.2%), micropapillary (14.3%), papillary (9.1%), lepidic (5.2%), invasive mucinous (3.9%) and colloid ADC (0.4%). Eighteen significant gene mutations were detected: *TP53* (46%), *KRAS* (33%), *KEAP1* (17%), *STK11* (17%), *EGFR* (14%), *NF1* (11%), *BRAF* (10%), *SETD2* (9%), *RBM10* (8%), *MGA* (8%), *MET* (7%), *ARID1A* (7%), *PIK3CA* (7%), *SMARCA4* (6%), *RB1* (4%), *CDKN2A* (4%), *U2AF1* (3%) and *RIT1* (2%). In the *RTK/RAS/RAF* signaling pathway, around 75% of the examined ADCs presented with driver gene mutations (*KRAS, EGFR, BRAF, ERBB2, MAP2K1, NRAS* and *HRAS*), gene fusions (*ROS1, ALK*, and *RET*), gene amplifications (*ERBB2* and *MET*) and exon skipping (*MET*). They also discovered *NF1* (*RTK/RAS/RAF* pathway suppressor gene, 8.3%) and *RIT1* (constitutes *RTK/RAS/RAF* pathway, 2.2%) mutations.

mRNA profiling subdivided ADC into three transcriptional subtypes: the terminal respiratory unit (TRU), the proximal-inflammatory (PI) and the proximal-proliferative (PP) mRNA subtypes [3]. The TRU subtype presented with frequent *EGFR* mutations and kinase fusions, while the PI subtype was characterized by co-mutations of *NF1* and *TP53*. The PP subtype was enriched with *KRAS* mutation and *STK11* inactivation. This clustering was partially overlapped by those observed in the protein expression profiles.

DNA methylation profiling also divided the ADC into three categories; CpG island methylator phenotype (CIMP)-high, CIMP-intermediate and CIMP-low subtypes [3]. CIMP-high tumors have frequent methylated *CDKN2A, GATA2, GATA5, HIC1, HOXA9, HOXD13, RASSF1, SFRP1, SOX17* and *WIF1*.

#### 2.2.3. Molecular Abnormalities and Histological Pattern in ADC

In the TCGA report, the PI subtype was characterized by a solid morphology. Here we summarize the studies which have demonstrated certain molecular alterations and morphological patterns.

*EGFR* mutation, the most common therapeutic targeted driver mutation in ADC, is associated with a micropapillary pattern [6]. Lepidic ADC (categorized as bronchioloalveolar carcinoma in the previous WHO classification) is also reported to be related to *EGFR* mutations [7,8,9].

*ALK* rearrangements are observed in approximately 4–5% of ADCs [10], and are characterized by the presence of signet ring cells forming an acinar structure with mucin production [11,12,13]. The morphological characteristics of *ROS1*- and *RET*-rearranged ADCs, which comprise 1% each of ADCs [10], are also reported to be similar to the *ALK*-rearranged ADCs [14]. Some reports have shown the association between *ROS1* fusions and psammomatous calcifications [15,16]. ADCs with *RET* fusions presented with poorly-differentiated histology when compared to those with *EGFR* mutations or *ALK* rearrangements [17].

Micro-RNAs are now considered as attractive targets of diagnostic and predicting markers. Nadal et al. performed clustering of 356 miRNAs, and identified three major clusters of lung ADCs that were correlated with the histologic subtype of lung ADC [18]. Cluster 1 included lepidic or mucinous invasive ADCs, while clusters 2 and 3 comprised acinar and solid tumors. Nineteen miRNAs were detected with solid pattern and 30 with lepidic pattern. Three miRNAs encoded at 14q32 (miR-411, miR-370 and miR-376a) were associated with poor survival.

The mucin-rich subtype including mucinous ADC (IMA) and colloid ADC (CA), is shown to harbor *KRAS* mutations more often than the non-mucinous subtype [19,20,21,22,23]. *NRG1* fusion genes have been observed in 13–27% of *KRAS*-wild type tumors [21,24]. *NTRK1*-rearrangement is also identified in IMA [25]. Other mutations, including *EGFR, STK11, TP53, CDKN2A, RB1, PIC3CA, APC, STK11, SMAD4, SMO, c-KIT and HFN1A,* have been detected by NGS analysis [20,26]. *KRAS* mutations have been observed along with *NKX2-1/TTF1* repression, and associated with mucinous carcinoma development [27] and Napsin A downregulation [28].

The most common genetic abnormality in enteric carcinomas (EC) was *KRAS* mutation followed by *EML4-ALK* fusion, *NRAS* mutations and *EGFR* mutations [29,30]. Moreover, four out of five enteric ADCs had mutations in mismatch-repair genes, and tumor mutational burden (TMB) levels were higher than those seen in control ADCs [29].

CDX2 and MUC2, the intestinal IHC markers frequently positive in EC, are reported to be expressed in CA [31]. Furthermore, IMA, CA and EC are occasionally assumed as tumors on the same spectrum [20,26,28]. A recent study attempted to reclassify these tumors according to the IHC status [26].

Fetal ADC (FA) is occasionally subdivided into low- and high-grade carcinomas according to the nuclear characteristics. Genetic abnormalities in the Wnt pathway and aberrant beta-catenin overexpression are observed due to *CTNNB1* mutation in low-grade FA [32]. A recent analysis with NGS showed *BRCA2* and *TSC2* mutations in FA [33]. High-grade FA, on the other hand, was characterized by p53 overexpression and mutations in both *EGFR* (20%) and *PIC3CA* (7%) [34].

### 2.3. Squamous Cell Carcinoma

#### 2.3.1. Morphological Subtypes

SQCs are divided into keratinizing, non-keratinizing, and basaloid types. Non-keratinizing SQC is sometimes difficult to distinguish from poorly-differentiated solid ADCs, and due to which, IHC analysis is warranted for diagnosis. Basaloid type SQC is also positive for the IHC markers of SQC, but consists of unique molecular profiles. The prognostic difference between each histological subtype is controversial [2].

#### 2.3.2. Molecular Abnormalities in SQC Confirmed by TCGA

In 2012, the TCGA project released the results of the molecular analysis for 178 SQC [4]; 360 exonic mutations, 165 genomic rearrangements, and 323 segments of copy number alteration per one SQC were observed on an average. This complex alteration is assumed to be caused by smoking. The significant genetic mutations observed in their study were *TP53, CDKN2A, PTEN, PIK3CA, KEAP1, MLL2, HLA-A, NFE2L2, NOTCH1, RB1* and *PDYN,* with nearly 90% of the tumors harboring *TP53* mutations. Mutations in the oxidative stress-related pathway (*KEAP1* and *NFE2L2*), squamous cell differentiation-related genes (*SOX2* and *TP6*3) and the *PI3K/RTK/RAS* pathway were seen in 34%, 44% and 69% cases, respectively; *CDKN2A* inactivation was noted in 72% of the cases.

The mRNA profiling categorized SQC into four subtypes; classical, basal, secretory and primitive. The classical subtype is characterized by alterations in *KEAP1*, *NFE2L2* and *PTEN* genes, hypermethylation and chromosome instability. The basal subtype presents with a high frequency of *NF1* mutations. The secretory subtype is characterized by *TP53* and *RB1* activation. The primitive subtype frequently harbors *RB1* and *PTEN* mutations.

Furthermore, SQC was categorized into four subtypes following DNA methylation and miRNA profiling, which overlapped, to some extent, with the mRNA profiling subtypes. For example, methylation cluster 4 (which shows little DNA hypermethylation) and miRNA cluster 3 contain most of the primitive subtypes defined by mRNA. The majority of the SQCs in cluster 3 comprise classical mRNA subtypes with *NFE2L2* mutations. The authors in TGCA report finally divided SQC into three categories based on integrative clustering of somatic mutations, DNA copy numbers, DNA methylation and mRNA expression data using the iCluster method.

## 3. Molecular Alterations as Diagnostic Markers between ADC and SQC

As mentioned earlier, ADC and SQC have distinct molecular profiles. The diagnostic values of these alterations have been evaluated in several studies; Shinmura et al. identified 6 and 24 genes specifically expressed in ADC and SQC, respectively, in the RNA-seq data of TCGA database [35]. CLCA2 was found as a specific IHC marker of SQC [35]. Sun et al. identified 778 genes and 7 miRNA that were differently expressed between ADC and SQC via bioinformatics analysis of TCGA data [36]. Common core transcriptional factors of the networks, which is composed with transcriptional factor, miRNA and gene, were shown between ADC and SQC. Meanwhile, miR-29b-3p was demonstrated to be upregulated only in ADC, whereas in SQC, miR-1, miR-105-5p and miR-193b-5p were altered. Campbell et al. utilized exome sequence and copy number profiles from tumor and normal tissue pairs [37], and revealed *PPP3CA, DOT1L* and *FTSJD1* mutations and *MIR21* amplification in ADC; alternatively, *RASA1* mutation and *MIR205* amplification were reported in SQC. Shoshan-Barmatz et al. demonstrated 23 potential proteins that were differently expressed in ADC and SQC, and detected via proteomics and RNA-seq data [38]. They proposed that nuclear staining of SMAC protein by IHC can be used as a diagnostic biomarker between ADC and SQC.

## 4. Neuroendocrine Tumor

### 4.1. Morphological Definition

Neuroendocrine tumors (NETs) include typical carcinoid (TC), atypical carcinoid (AC), small cell carcinoma (SCC), and large cell neuroendocrine carcinoma (LCNEC). TC and AC are considered as low-grade NET, while SCC and LCNEC are regarded as high-grade NET. The morphological features including necrosis and Ki-67 index (widely used cutoff value, 4–5% [39]) divides low-grade NETs into TC and AC. All of the NETs demonstrate neuroendocrine differentiation, which is confirmed with IHC markers (synaptophysin, chromogranin A and CD56). SCC and LCNEC are discriminated based on morphological features, such as the size of the tumor cell, prominent nucleoli in LCNEC, and amount of cytoplasm. However, inter-observer variability is generally high, particularly in biopsy specimens due to the amount of tissue available and the presence of crush artifacts [40,41]. Some IHC markers (*BAI3, CDX2* and *VIL1*) distinguishing SCC and LCNEC have been studied, but they are not widely used in daily practice [42].

### 4.2. Molecular Abnormalities

#### 4.2.1. Typical and Atypical Carcinoids (Low-Grade NET)

An NGS analysis targeting 48 genes showed non-activating mutations in *EGFR* (6%, TC; 12%, AC), *ERBB2* (12%, TC and AC), *RET* (6%, TC and AC), *MET* (12%, AC)*, KIT* (12%, AC), *KRAS* (6%, AC)*, KDR* (6%, AC) and *FGFR1* mutations (6%, AC) [43]. In another NGS study, only one out of 25 TC and AC cases harbored *BRAF, SMAD4, PIK3CA* and *KRAS* mutations in the tissues. These findings indicate that common mutations in NSCLC are relatively rare in low-grade NET [44].

PCR-based microRNA analysis of TC and AC with lymph node metastases demonstrated the presence of 24 miRNAs that were differently regulated between TC and AC [45]. Among them, miR-129-5p, miR-409-3p, miR-409-5p, miR-185 and miR-497 were significantly upregulated in TC. The expressions levels of the 29 miRNAs were different between the metastatic and non-metastatic cases indicating that miRNA profile may be used as a predictive biomarker.

#### 4.2.2. SCC and LCNEC (High-Grade NET)

##### Molecular Abnormalities of SCC Confirmed by TCGA

The TCGA project showed extremely high frequencies of bi-allelic inactivation of both *TP53* (98%) and *RB1* (91%) in 110 SCC [5]. Alternative deregulation pathway of *RB1* was also observed as cyclin D1 upregulation coded in the *CCND1* gene (2%). *Notch* family genes were found to be inactivated in 25% of SCC, leading to neuroendocrine differentiation. In addition, oncogenic *TP73* rearrangement (13%) was discovered. Other gene alterations including *CREBBP* (15%), *EP300* (13%), *MYCL1* (9%), *PTEN* (9%), *MYC* (6%), *KIT* (6%), *FGFR1* (6%), *MYCN* (4%) and *PIK3CA* (3%) have been noted [5]. SCC was further divided into two clusters based on the expression analysis; the majority (77%) were characterized by highly-expressed *CHGA* (chromogranin A), *GRP* (gastrin releasing peptide), *DLK1* (an inhibitor of *Notch* signaling) and *ASCL1* (whose expression is inhibited by active *Notch* signaling) [5].

##### Molecular Alterations of LCNEC

It is implied that LCNEC is made up of subgroups that resemble SCC and NSCLC by their molecular profiles. Rekhtman et al. performed an NGS analysis in 45 LCNEC and demonstrated alterations in *TP53* (78%), *RB1* (38%), *STK11* (33%), *KEAP1* (31%) and *KRAS* (22%) [46]. They identified two major and one minor subsets: SCC-like LCNEC, characterized by *TP53* + *RB1* co-mutation/loss and MYCL amplification; NSCLC-like LCNEC, characterized by the lack of co-altered *TP53* + *RB1* and almost universal occurrence of NSCLC-type mutations (*STK11, KRAS* and *KEAP1)*; and carcinoid-like LCNEC, characterized by *MEN1* mutations and low mutational burden. Mitotic activity in the SCC-like subset is significantly higher than that in the NSCLC-like subset, with different cytomorphological features. Alterations in the *Notch* family genes were observed in 28% of the NSCLC-like subtype indicating neuroendocrine differentiation. Miyoshi et al. also showed consistent profiles of LCNEC with TP53 (71%) and RB1 (26%) mutations [47]. Additionally, *PIK3CA/AKT/mTOR* pathway alterations were reported in 15% of the LCNEC. Notably, LCNEC combined with NSCLC shared the same driver mutations previously reported in NSCLC [47]. Thus, these molecular alterations in LCNEC may be used as targets.

##### Transformed SCC from NSCLC after Treatment with EGFR-Tyrosine Kinase Inhibitors (TKI)

*EGFR* mutation-positive ADCs treated with tyrosine kinase inhibitors occasionally acquire resistance, and 15% of them transform into SCC [48]. A TKI-resistant NSCLC cell line demonstrated *RB1* inactivation associated with SCC transformation [49]. Sequencing of SCC-transformed NSCLC has revealed high proportions of inactivated *RB1* and *TP53* [50]. Moreover, these mutations are suggested to have occurred before TKI treatment; therefore, the identification of these mutations in advance may predict the risk of resistance to TKI [51]. In another study, alterations in the *Notch-ASCL1* signaling pathway appeared to play a role during the early phase of secondary SCC induction [52].

#### 4.2.3. Genetic Differences Between Low- and High-Grade NET

Recent studies compared the molecular profiles between low-grade (TC and AC) and high-grade (SCC and LCNEC) NET. *MEN1, PSIP1, ARID1A* and *EIF1AX* mutations were common in low-grade NET whereas, alterations in *TP53, RB1* and genes belonging to the *PI3K/AKT/mTOR* pathway were associated with high-grade NET [52,53].

## 5. Other Histological Subtypes

### 5.1. Adenosquamous Carcinoma

Adenosquamous carcinoma (ADSQ) histologically consists of an ADC and a squamous cell carcinoma component. A study using microdissection showed that most genetic mutations were common in both the components, whereas some mutations, including *KRAS, HER2* and *EGFR*, were unique in the ADC component [54]. *EGFR* mutation is less common in ADSQ with a solid ADC component when compared to those with well-differentiated ADC components; *ALK* or *RET* fusion is more frequent in the former type of ADSQ [54].

### 5.2. Sarcomatoid Carcinoma

Sarcomatoid carcinoma includes pleomorphic carcinoma (which consists of an epithelial and a sarcomatoid component), spindle cell carcinoma and giant cell carcinoma. Fallet et al. showed *KRAS* (27.2%), *EGFR* (22.2%), *TP53* (22.2%), *STK11* (7.4%), *NOTCH1* (4.9%), *NRAS* (4.9%) and *PI3KCA* (4.9%) mutations in 81 patients with sarcomatoid carcinoma [55] Terra et al. also identified *AKT1*, *JAK3* and *BRAF* mutations in this morphological subtype [56]. In addition, *ALK* rearrangement has been identified in one case [56]. Notably, *EGFR* mutations in this carcinoma were almost always “rare mutations” (such as exon 2, 18 or 20) [55,56]. The origin of the cell types in sarcomatoid carcinoma is suggested to be the same because the same mutations were shared in common between the epithelial and sarcomatoid components [55,57].

### 5.3. Salivary Gland-Type Tumor

The salivary gland-type tumor comprises mucoepidermoid and adenoid cystic carcinomas with morphological features similar to those of salivary gland tumors. Approximately 50% to 100% of pulmonary mucoepidermoid carcinomas harbor *MAML2* translocations, like the salivary gland tumors [58,59,60,61]. Genetic alterations in *HER2, EGFR* and *KRAS* have also been reported in pulmonary mucoepidermoid carcinoma [62,63,64,65,66]. However, *EGFR, KIT, KRAS, BRAF, ALK, PIC3CA, PDGFRA* and *DDR2* mutations were not detected in small-scale studies targeting adenoid cystic carcinoma [67,68,69].

### 5.4. Lymphoepithelioma-Like Carcinoma

Lymphoepithelioma-like carcinoma (LELC) is an undifferentiated tumor characterized by the presence of large cells with indistinct cell borders and prominent infiltrating lymphocytes. Ebstein-Barr Virus infection is associated with LELC, and *EGFR* mutations have been reported in eight (18%) out of 46 cases [70]; on the other hand, no *EML4-ALK* translocation or *KRAS* mutations were reported in these tumors [71].

### 5.5. NUT Carcinoma

NUT carcinoma (NC) is defined as a carcinoma with *NUT* gene rearrangement. The most common pattern of fusion is observed between *NUT* and *BRD4*, followed by that between NUT and *BRD3* [72]. Although the morphological features of NC are not included in the definition, it is characterized by nests of monomorphic, small- to intermediate-sized undifferentiated tumor cells [73]. In some instances, the diagnosis of NC is confirmed using NGS platforms indicating its potential for the confirmation of rare tumors in clinical practice [74,75].

## 6. Targeted Treatment

Recently, molecular subtyping of lung cancer has led to the approval and use of molecular targeted therapies. The targeted therapies are listed in Table 1.

### 6.1. Targets and Therapies for Non-Small Cell Lung Cancer

#### 6.1.1. EGFR Mutations

Somatic mutations in EGFR are the most common types of mutations seen in patients with NSCLC, and the discovery of EGFR-TKIs has become a pioneer in the era of targeted therapy. Approximately 30% of all advanced NSCLC cases have been reported in Asians and 20% in Caucasians [76]. Gefitinib, a first-generation EGFR-TKI, has shown superior progression free survival (PFS) in patients harboring *EGFR* mutations when compared to standard chemotherapy in EGFR-mutated advanced NSCLC patients [77,78]. Erlotinib, another first-generation EGFR-TKI, also showed longer PFS than standard chemotherapy against major *EGFR* mutations (L858R, del19), and has been widely used as a first-line treatment [79,80]. In 2013, afatinib, a second-generation EGFR-TKI, which acts as an inhibitor of all four members of the ERBB family, was approved for treating advanced NSCLC harboring EGFR mutations. A significant improvement in PFS and overall survival was noted with afatinib when compared to platinum doublets chemotherapy for EGFR mutation-positive NSCLC patients [81,82,83]. Furthermore, afatinib and gefitinib were compared in a phase II study wherein, median PFS was found to be significantly longer in Asian patients treated with afatinib [84]. Another second-generation EGFR-TKI, dacomitinib, demonstrated longer PFS when compared to gefitinib during the late phase in the EGFR-mutated population, except in patients with brain metastasis [85].

Patients with EGFR mutation can develop resistance to first and second-generation EGFR-TKI, which is often mediated by the T790M resistance mutation. T790M is an acquired resistance mechanism found in over half of the patients. Osimertinib, a third-generation irreversible EGFR-TKI, is selective for EGFR-TKI sensitizing and T790M resistance mutations. In a phase II trial comparing osimertinib to platinum doublet therapy among patients with T790M mutation undergoing first-line TKI therapy, the median PFS was significantly longer in those receiving osimertinib [86]. Recently, another phase III trial, which compared osimertinib with standard EGFR-TKIs in patients with previously untreated advanced NSCLC and EGFR mutations revealed significantly longer median PFS in patients receiving osimertinib when compared to those undergoing standard EGFR-TKIs therapy (18.9 months vs. 10.2 months) [87]. Before receiving approval for the use of osimertinib as a first-line treatment, a rebiopsy of the tissue was required. Liquid biopsy was often performed in cases where rebiopsy could not be performed; however, the efficacy of osimertinib in patients with T790M mutations detected by liquid biopsy is unclear and needs to be evaluated in future. Moreover, the efficacy and selection of EGFR-TKI after developing resistance against osimertinib has not been reported so far. EGFR C797S mutations accounts for 7% [88] of the acquired resistance, and treating with a combination of first-generation EGFR-TKIs and osimertinib showed efficacy [89]. Other acquired resistance after using osimertinib consists of uncommon EGFR mutations, HER amplification (2%), HER2 mutation (1%), SPTBN1-ALK (1%), MET amplification (15%), BRAF mutation V600E (3%), KRAS mutations (G12D/C, A146T, 3%) and PI3KCA mutations (7%) [88]. This means cases with acquired resistance to osimertinib would not be sensitive for EGFR-TKI re-challenge.

Combination therapy with chemotherapy or anti-VEGF agents has emerged as a promising strategy. NEJ009, the first phase III study, showed a significantly prolonged PFS (20.9 months) in patients receiving a combination of EGFR-TKI and platinum doublet chemotherapy when compared to those treated with gefitinib alone [90].

The most common EGFR mutations, exon 21 L851R and exon 19 deletions, respond to first, second, and third-generation EGFR-TKI. However, there is limited data concerning uncommon EGFR mutations, such as exon 20 insertions, exon 18 point mutations, and complex mutations, which account for 10% of all EGFR mutations and are associated with poor prognosis and survival. Afatinib has proven effective in patients harboring certain types of uncommon EGFR mutations, especially G719X, L861Q and S768I, but is less active in other mutations types [91]. Potizotinib, TAK-788 and TAS6417 are known to have clinical activity for exon 20 insertions [92]; however, they have not been approved by the FDA so far. Structural alternation induced by Exon 20 insertions of EGFR makes it difficult to combine with conventional EGFR kinase inhibitor, so the other drug improvement is under way.

#### 6.1.2. ALK Rearrangements

ALK rearrangements were initially discovered in 2007 by Soda and colleagues as the driver oncogene in solid tumors [93]. An early phase clinical trial demonstrated the clinical efficacy of crizotinib (PF-02341066), which was initially developed as a MET inhibitor [94]. Crizotinib was approved as the first-in-class ALK inhibitor for ALK-rearrangement-positive NSCLCs (comprising ~5% of the ADCs). In the PROFILE trial series, crizotinib demonstrated superiority over platinum doublets chemotherapy on PFS in patients with previously untreated and treated advanced NSCLC harboring ALK translocation [95,96]. One of the limitations of these studies is that, crizotinib is not an ALK-specific inhibitor. Among the second-generation ALK inhibitors, alectinib has high selectivity for ALK rearrangement and can overcome L1196M or C1156Y, the major secondary mutations that lead to resistance to crizotinib. Phase III trials such as ALEX [97], J-ALEX [98] and ALESIA [99], have demonstrated the superiority of alectinib over crizotinib in terms of PFS and safety in the first-line setting. Thus, alectinib has become the concrete standard of treatment for previously untreated advanced NSCLC patients with ALK rearrangements. Brigatinib, ceritinib and loratinib also have high selectivity for ALK inhibition, and have the potential to overcome resistance via secondary mutations in *ALK*. However, in terms of the toxicity profile, alectinib continues to be the standard treatment for now.

With the emergence of various ALK inhibitors, sequential use of ALK inhibitors to overcome acquired resistance has become a treatment choice, and the selection of ALK inhibitor has become more important than ever. Rebiopsy is necessary for choosing consequent agents after the failure of the preceding agent. On the other hand, liquid biopsy with NGS may prevail over tissue-based gene analysis. A study, which compared tissue with sequenced cell-free DNA from plasma, reported that *ALK* rearrangements detected in tissues were also detected in plasma in 79% of the cases [100]. Thus, less-invasive sampling techniques such as liquid biopsy could broaden the spectrum of patients who can receive benefits from sequential treatment with ALK inhibitors.

The acquired resistant mechanism in ALK fusions for ALK inhibitor is different from that in EGFR-TKI with sensitivity for tyrosine kinase inhibitors; hence, a repeatable oral kinase inhibitor strategy can be assumed. However, there is no evidence of treatment choice after developing resistance against second-generation ALK inhibitor, and further validation is needed in the clinical trial.

#### 6.1.3. ROS1 Rearrangements

*ROS1* rearrangements are less common than ALK, and account for 1% of the NSCLCs. ROS1 has high homology with ALK comprising 70% of the amino acids. Therefore, the key drugs for treatment are also similar to the ALK inhibitors. Crizotinib has shown high response rates against *ROS1* rearrangement NSCLC [101,102], and is the first choice for treatment. In addition, the effects of other ALK and ROS/TRK inhibitors such as Lorlatinib [103] and Entrectinib [104] are under investigation. *ROS1* mutation, *EGFR* activation and epithelial-to-mesenchymal transitions are implied in drug resistance [105]. Cabozantinib is suggested to have the potential to overcome acquired resistance after crizotinib treatment [106].

#### 6.1.4. BRAF Mutations

*BRAF*^V600E^ mutation is observed in 1–2% of advanced NSCLCs [107] and 85% of them are ADC. Associations between *BRAF* mutation and gender, smoking, histology and stage of cancer are poorly understood, indicating that patients with *BRAF* mutations in NSCLC have heterogeneous clinical backgrounds. Dabrafenib, a tyrosine kinase inhibitor that targets *BRAF* mutations, has shown 30% clinical activity in patients with NSCLC [108]. The combination of *BRAF* and *MEK* inhibitors has shown an even higher response; combined treatment with Dabrafenib (*BRAF* inhibitor) and Trametinib (*MEK* inhibitor) demonstrated clinical effectiveness against previously untreated and treated NSCLC. In previously untreated patients with *BRAF^V600E^* mutation, the response rate was 64% (95% CI, 46–79%) and PFS was 10.9 months (95% CI, 7.0–16.6 months) [109] whereas, in previously treated NSCLC with *BRAF^V600E^* mutations, the response rate was 63.2% (95% CI, 49.3–75.6%) and PFS was 9.7 months (95% CI, 5.7–13.6 months) [110].

#### 6.1.5. Other Genetic Targets and Reagents for NSCLC

Other target driver oncogenes, besides *EGFR*, *ALK*, *ROS1* and *BRAF,* have also been characterized in NSCLC.

*RET* rearrangements are found in 1–2% of NSCLCs [110]. They are commonly found in younger patients below the age of 60, non-smokers or former light smokers, and patients with poorly-differentiated ADC [17]. Multi-kinase inhibitors such as Cabozantinib and Lenvatinib have shown 28% [110] and 16% [111] overall response rates, respectively. Vandetanib, a multiple kinase inhibitor, has shown an overall response of 53% with 4.7 months of PFS in the LURET study [112]. Recently, LOXO-292, a potent and highly selective RET inhibitor based on the kinome model, demonstrated an impressive overall response rate of 65% in RET fusion-positive NSCLC patients in a phase I study [113].

*MET* exon14 skipping mutation (*MET*Δex14) occurs in 2–4% of ADCs and is rather frequently found in elderly patients when compared to those with NSCLC harboring *EGFR* or *KRAS* mutations and non-smokers [114]. *MET* amplifications also occur in 3–5% of *EGFR*-wild type ADCs [115]. The effectiveness of Crizotinib has been clinically shown in patients with *MET* amplification or *MET* Δex14 [114,116,117,118]. Capmatinib, a new agent that targets *MET*, demonstrated clinical benefits with overall response rates of 39.1% and 79.0% in previously treated and untreated *MET* Δex14 NSCLC patients, respectively [115].

NTRK fusion-positive NSCLC is estimated to account for less than 1% of the NSCLC patients [119]. Several studies have demonstrated that TRK fusion proteins promote oncogenesis by mediating constitutive cell proliferation and survival, and larotrectinib and entrectinib have emerged as effective TKI inhibitors [120].

### 6.2. Biomarkers of Immunotherapy for NSCLC without Targetable Gene Alterations

Immune checkpoint blockade (ICB) consolidated its major role in the treatment of NSCLC patients without targetable genetic alterations and extended its use by showing impressive data on ICB combination therapies (e.g. combined with chemotherapy). The use of predictive biomarkers for ICB therapy, such as programmed death-ligand 1 (PD-L1) expression and TMB testing, has gained popularity. In addition, blood-based tests including soluble PD-L1 have been proposed to have the potential for treatment, monitoring and prognosis prediction [121]. Multimodal approaches such as combining biomarkers analysis from non-solid biological tissue may become effective.

No major studies concerning new genetic alterations or innovative targets have been conducted so far. Therefore, combination therapy, such as ICB plus chemotherapy, is another approach to increase the efficacy besides finding predictive biomarkers. There are benefits using the combinational approach; however, further validation including toxicity is needed.

### 6.3. Targets and Therapies for NET

Delta-like protein (DLL3), a non-activating ligand of Notch, is known to be expressed in 80% of SCC [122]. DLL3 is a promising target because its expression is generally restricted, except for the brain tissue in the physiological state. Rovalpituzumab tesirine, a humanized monoclonal antibody against DLL3 (rovalpituzumab) conjugated with a pyrrolobenzodiazepine-type cytotoxic reagent, showed a response rate of 38% in patients with >50% DLL3 expression [122].

## 7. Future Perspectives

Results of large-scale comprehensive molecular analyses including the TCGA project have demonstrated the recent developments in new targeted therapies. Gene sequences are becoming a crucial part of clinical practice along with NGS platforms, which are widely available. The detection of a driver mutation in a patient does not suggest that he/she can receive precision medicine owing to factors such as the lack of therapy or rapid progression of the disease. Less than 30% of lung cancer patients detected with gene alterations in *ROS1* and *RET* are reported to be able to receive targeted therapies [123]. Prompt treatment after diagnosis along with reduction in turn-around time and invention of new drugs are strongly warranted.

There is a huge amount of data available currently due to the possibility of collecting information by NGS. According to a research by the National Institutes of Health (NIH), 76% of the oncologists in the USA use NGS, and 52.4% have reported that they have altered their management therapies based on the results obtained from the NGS. However, there is no prospective trial data demonstrating the clinical benefits of broad NGS testing compared with limited molecular testing only for approved therapies. In order to organize large datasets and utilize them for clinical use, it is necessary to combine them with real world data and build an IT platform.

Recent innovations in imaging analysis and artificial intelligence (AI) are leading the classical classifications based on histology onto the next stage. Genetic alterations and prognosis are predicted by histological images using the deep-learning method in lung cancer [124,125]. Data analysis, including those from the TCGA project, have partially made it possible to reproduce morphological classifications and propose brand-new classifications based on molecular profiles [36]. Moreover, enormous data obtained from broad sequencing in daily practice would be a treasure-trove for new drug development and investigation. The costs of these molecular analyses are relatively high when compared to conventional histological analysis using hematoxylin-eosin staining of formalin-fixed, paraffin-embedded tissues; however, combinations with other perspectives from AI and molecular profiles may prove beneficial in lung cancer.

## 8. Conclusions

Next-generation sequencing has opened the door of totally novel clinical practice and investigation for lung cancer. As more targeted therapy becomes available, broad genetic analysis would be a part of daily pathological diagnosis in the future. The knowledge obtained from previous studies should be understood as a foundation for not only researchers of pathophysiology and therapy, but also for pathologists and oncologists who participate in next-generation clinical practice.

## Figures and Tables

**Figure 1 cancers-11-00599-f001:**
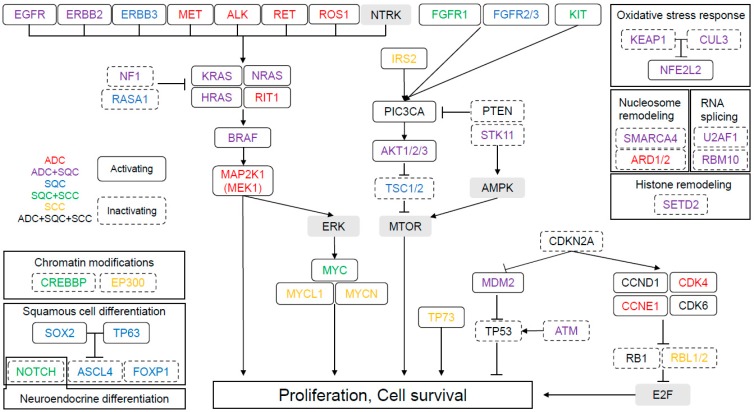
Summary of genetic mutations in lung adenocarcinoma, squamous cell carcinoma, and small cell carcinoma detected by The Cancer Genome Atlas project. [3,4,5].

**Table 1 cancers-11-00599-t001:** Targeted therapies.

Histology	Targeted Molecule	Reagents	FDA-Approved (April 2019)	Note
**NSCLC**	EGFR (ADC 14%, SQC 9%)	Gefitinib (First-generation)	Yes	Selective and reversible TKI.
		Erlotinib (First-generation)	Yes	Selective and reversible TKI.
		Afatinib (Second-generation)	Yes	Irreversible ErbB family blocker.
		Dacomitinib (Second-generation)	Yes	Irreversible TKI.
		Osimertinib (Third-generation)	Yes	Irreversible and also active against the resistance mutation (T790M).
		Poziotinib	No	Irreversible and active against exon20 mutation and HER2 mutation.
		TAK-788	No	Active against exon20 mutation and HER2 mutation.
		TAS6417	No	Selective against exon 20 insertion mutation.
	ALK (ADC 5%)	Crizotinib(First-generation)	Yes	Multi-targeted TKI.
		Alectinib (Second-generation)	Yes	Highly selective inhibitor for ALK.
		Ceritinib (Second-generation)	Yes	Highly selective inhibitor for ALK.
		Brigatinib (Second-generation)	Yes	ALK/ROS1 inhibitor.
		Lorlatinib (Third-generation)	Yes	ALK/ROS1 inhibitor.
	ROS1 (ADC 1%)	Crizotinib	Yes	
		Lorlatinib	No	
		Entrectinib	No	Inhibits ROS1, TRK and ALK.
	BRAF (ADC 7%, SQC 4%)	Dabrafenib/trametinib	Yes	Reversible ATP-competitive kinase inhibitor.
	RET (ADC 1%)	Cabozantinib	No	Multi-targeted TKI.
		Lenvatinib	No	Multi-targeted TKI.
		Vandetanib	Yes	Multi-targeted TKI.
		LOXO-292	No	Selective RET inhibitor
	MET (ADC 2–4%)	Crizotinib	No	
		Capmatinib	No	Reversible MET inhibitor.
		Tepotinib	No	Reversible MET inhibitor.
	NTRK (<1%)	Entrectinib	No	
		Larotrectinib	Yes	TRK inhibitor.
	PD-1	Nivolumab	Yes	IHC: 28-8
		Pembrolizumab	Yes	IHC: 22C3
	PD-L1	Atezolizumab	Yes	IHC: SP142
		Duvalumab	Yes	IHC: SP263
**SCC**	DLL3 (80%)	Rovalpituzumab tesirine	No	Antibody against DLL3 conjugated with cytotoxic reagent.

NSCLC, non-small cell lung cancer; SQC, Squamous cell carcinoma; EGFR, epidermal growth factor receptor; ADC, adenocarcinoma; TKI, tyrosine kinase inhibitor.

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
