# Peer review of "Molecular and Morphological Profiling of Lung Cancer: A Foundation for “Next-Generation” Pathologists and Oncologists"

_cancers, 2019, doi:10.3390/cancers11050599_

Round 1
Reviewer 1 Report
While this is a nice summary of the molecular and morphological profiling of lung cancer, as well as the current targeted therapies, I found this review lacks certain degree of focus, and missing the thoughtful part regarding how to incorporate these profiles into the "foundation for next-generation pathologists and oncologists" as they claimed.
Some minor issues:
78-87: need reference
252-253: stated EGFR 22.2%, but 255-256 says “EGFR mutations were almost always rare in this carcinoma”. Please clarify.
Author Response
Comments and Suggestions for Authors
While this is a nice summary of the molecular and morphological profiling of lung cancer, as well as the current targeted therapies, I found this review lacks certain degree of focus, and missing the thoughtful part regarding how to incorporate these profiles into the "foundation for next-generation pathologists and oncologists" as they claimed.
- Existing knowledges including what we presented in the manuscript would be the basement for investigation and may be a foundation of clinical practice in the future. We have added the last paragraph to describe the meaning of the summarized information.
Some minor issues:
1. 78-87: need reference
- We have added the reference.
2. 252-253: stated EGFR 22.2%, but 255-256 says “EGFR mutations were almost always rare in this carcinoma”. Please clarify.
- We have clarified the sentence.

Reviewer 2 Report
Well-written, focussed, and informative review. No major revision suggested, suitable for publication.
Author Response
Well-written, focussed, and informative review. No major revision suggested, suitable for publication.
- We appreciate your reviewing our manuscript.
Reviewer 3 Report
The review article "Molecular and morphological profiling of lung cancer: a foundation for "next-generation" pathologists and oncologist by Kashima et al. summarizes the current knowledge about histological characteristics, molecular profiles and therapeutic options for the different subsets of lung cancers. The review is well structured and comprehensively details the different topics with an adequate amount of cited literature.
Minor comments:
- please revise english spelling
- Change LKB1 to STK11 on page 3, row 111 for consistency (its the same gene)
Author Response
The review article "Molecular and morphological profiling of lung cancer: a foundation for "next-generation" pathologists and oncologist by Kashima et al. summarizes the current knowledge about histological characteristics, molecular profiles and therapeutic options for the different subsets of lung cancers. The review is well structured and comprehensively details the different topics with an adequate amount of cited literature.
Minor comments:
1. please revise english spelling.
- We have corrected some typos.
2. Change LKB1 to STK11 on page 3, row 111 for consistency (its the same gene)
- We have corrected the sentence.
Reviewer 4 Report
The review by Kashima et al. can be accepted after following minor correction
1. Line 50. Some therapeutic reagents cannot be applied to squamous cell carcinoma. Please mention why?
2. Line 62 typo error. Micropapillary
3. Line 118. Were was weird sentence. “tumor mutational burden (TMB) levels 117 were was higher than those seen in control ADCs”
Author Response
The review by Kashima et al. can be accepted after following minor correction
1. Line 50. Some therapeutic reagents cannot be applied to squamous cell carcinoma. Please mention why?
We have added the reason.
2. Line 62 typo error. Micropapillary
We have corrected the typo.
3. Line 118. Were was weird sentence. “tumor mutational burden (TMB) levels 117 were was higher than those seen in control ADCs”
We have corrected the sentence.
Reviewer 5 Report
However, the idea of the study seems to be interesting, the summarized results of the project supported by NGS approach is disputable. Firtstly, access to advanced tools such as NGS i is limited in wide range of institutions, moreover, even these are able to be used the interpretation of the results is difficult. Second, these data are obtained rather from studies included low nubber of heterogenous samples. Even the NGS findings are promising, these usually do not agree with results obtained by simpler methods. Summarizing, the collected data and created atlas cannot be a guideliness for clinical procedures because the bias of reults is significant.
Author Response
Comments and Suggestions for Authors
However, the idea of the study seems to be interesting, the summarized results of the project supported by NGS approach is disputable. Firtstly, access to advanced tools such as NGS i is limited in wide range of institutions, moreover, even these are able to be used the interpretation of the results is difficult. Second, these data are obtained rather from studies included low nubber of heterogenous samples. Even the NGS findings are promising, these usually do not agree with results obtained by simpler methods. Summarizing, the collected data and created atlas cannot be a guideliness for clinical procedures because the bias of reults is significant.
- We do not assume that the results of previous studies we summarized in this article would directly be a guideline for future precision medicine. Meanwhile, existing knowledges including what we presented in the manuscript would be the basement for investigation and may be a foundation of clinical practice in the future. We have added the last paragraph to describe the meaning of the summarized information.
Round 2
Reviewer 1 Report
Thanks for the revision.
Reviewer 5 Report
Authors responded to reviewer comment.